## Thinking for Speaking in Context Rather than a Vacuum – A Path Towards Experimental Studies

The fundamental idea of Slobin's thinking-for-speaking hypothesis (1987) is that we create conceptualizations of the world (in the sense of Levelt 1989) in a way that lends itself to easy expression in our language/s. Despite the seemingly straightforward name of the hypothesis, Slobin (2003) specifies that it does not only pertain to linguistic production, but that it also more broadly covers other mental processes such as thinking for understanding and remembering. Many studies have found effects relevant to Slobin's hypothesis (see Sloupová 2024, pp. 16–29), but they relate it solely to the process of preparing for verbalization (i.e., thinking for *speaking* in the narrow sense), which is what supposedly shapes our preceding conceptualizations. None of these studies, notably, take into consideration the overall linguistic context of the verbalization event – specifically, the possible pre-activation of language and its constructions by preceding context of the production. If we suppose that processing this contextual 'input' is the first, preparatory part of the production process, we may find that it also plays a role in the structure of our conceptualizations.

We proposed a way of incorporating linguistic context into already existing methodologies, specifically following studies of linguistic effects on patterns of visual attention, also nicknamed 'seeing for speaking' (Carroll et al. 2004). We conducted an exploratory experiment focused on the effects of grammatical aspect on visual attention in motion events, an established area of seeing-for-speaking research (e.g., von Stutterheim et al. 2012; Flecken et al. 2014; Andresen et al. 2024). Nineteen Czech-English early bilinguals were tasked to describe short videos (see Fig. 1) while, per our addition, listening to unrelated background audio at the same time. Their gaze was monitored with an Eye-Link 1000+ eye-tracker. Each participant went through three blocks: the first two had a congruent language of perception and production (Czech and English), and the last block had incongruent languages, with English as the background language of perception and Czech as the language of production.

In our first, descriptive analyses, we looked into visual attention to the moving entity and endpoint of motion, which is connected to the presence of the progressive in the language of operation (von Stutterheim et al. 2012). We aimed to compare visual attention in the last, mixed block to the first two monolingual blocks and look for similarity to either of them (i.e., to the language of production or perception/context), but we see no significant differences between the first two blocks, and hence can make no claims on the effects of context. This is potentially due to high cognitive demands of the task, mixed conceptualizations in bilinguals (see Sloupová 2024, pp. 24–29) or the low number of participants. Even if this experiment does not end up showing significant effects, it serves as a first step towards incorporating linguistic context into thinking-for-speaking and seeing-for-speaking research. It allows us to discuss the difficulties of operationalizing linguistic context using examples from our design, and how currently used methodologies could be altered and built upon to account for it.

**Key words:** thinking for speaking, seeing for speaking, aspect, conceptualization, bilingualism

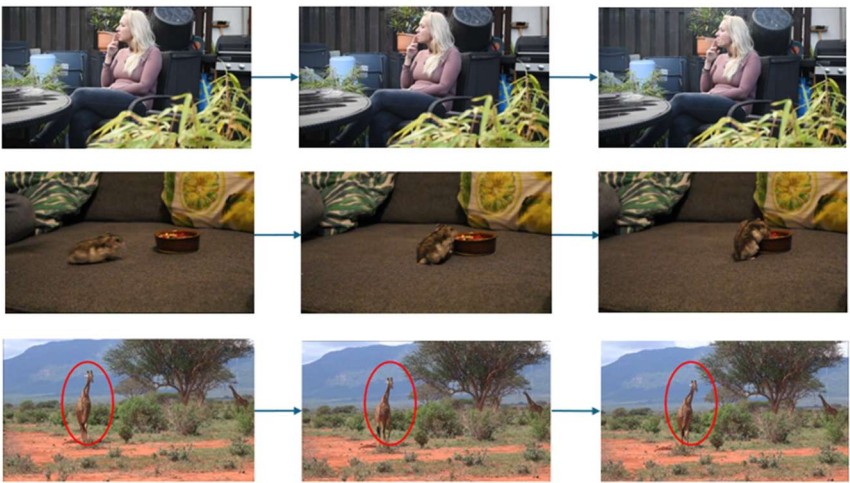

*Figure 1*. Example of video clips at three time points across their 2 second duration. Row 1: a filler clip (a woman smoking), row 2: a motion-event filler with target reached (a hamster running towards a bowl of food), row 3: critical stimulus (a giraffe walking towards a tree and another giraffe).

.

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
