# OpenReview forum: "Thinking for Speaking in Context Rather than a Vacuum – A Path Towards Experimental Studies"
_CUNI.cz/2024/CJOLPhD — CUNI 2024 CJOLPhD Submission_

### Official Review · ~Maria_Onoeva1 · 2025-01-07
**Good job!**

As someone who’s not super familiar with the theoretical foundations of the study, I really appreciated the background you provided earlier. But to be honest, I’m not entirely sure I understood the research questions or hypotheses for this study. I suggest describing them a bit more precisely in the future. Also I noticed that in studies like this, it’s common to include basic demographic details about participants, mention where the study took place and provide statistical results even if they are not significant.

I have a few general questions, and maybe you could point me toward some relevant literature?

1) Why didn’t you test the second incongruent condition, where participants would hear Czech audio but produce English?
2) From what I understood, you only analyzed eye movements, am I right? What about analyzing speech production? I’m not exactly sure how you’d do that, but it seems like it would be crucial for testing linguistic pre-activation.
3) I’m really curious about linguistic pre-activation. Did someone test non-bilingual participants in the same way? I’d be interested to see results from people who don’t speak English (or any other foreign language) but still listen to it. Could that still count as linguistic pre-activation, since it’s a language, even if they don’t speak it? And how would their results compare to people who listened to random, non-linguistic noise?

---

### Official Review · ~Anna_Staňková1 · 2025-01-07
**Great!**

I think this is a well-structured abstract. In the first paragraph you write that "Many studies have found effects relevant to Slobin’s hypothesis" and then you cite a diploma thesis with an overview of the previous research. In this case I think it would be better to pick around 3 most influential studies from the overview and cite them directly. I would suggest adding some information about the statistical method you used and provide p-values  when you are writing about the results not being statistically significant.

---

### Official Review · ~Matúš_Godál2 · 2025-01-08

This abstract presents an experiment with the focus on the role of linguistic context in the conceptualization process as described by Slobin. The topic itself seems promising as it addresses the gap in the current research by incorporating contextual linguistic input into the design of experimental studies. The abstract provides the readership with a detailed theoretical background by ancoring it within an already established framework. Although this is a strength of the abstract, clear statement of the research questions or hypotheses is missing. Also, the text of the abstract in its entirety would benefit from its structuring (e.g. IMRaD format), facilitating better cohesion of the text. In addition to that, the provision of more information about the studied population would be appreciated; the readership is only informed about the number of the bilinguals who took part in the experiment. In regard to the presentation of the results, tbe abstract does not mention any statistical results that would strenghten the credibility of the findings even though no significant differences between the monolingual and mixed-language blocks were detected by the author.

---

### Official Review · ~Radek_Šimík1 · 2025-01-08
**Very nice abstract with a clear structure, just minor notes below**

- I really like the the intro - jumping right into the issue.
- I think the adverb "notably" is superfluous. In such a short abstract, everything should be notable. Also, you might want to state already at this point that this is what you are going to do and add that nobody has done it before.
- A minor terminological note: "grammatical aspect" might be a bit misleading (garden-path-y), as it could be understand as "an aspect/issue of grammar". Maybe "verbal aspect" or "the category of aspect" are clearer.
- A related note on contents: Since Czech and English are so different in terms of how they express aspectual properties of events, it remains unclear what is meant by "aspect" here.
- "listening to unrelated background audio" - for some reason I imagined music or something, saying that it's a linguistic audio stimulus would be beneficial.
- "In our first, descriptive analyses" seems superfluous.
- "the effects of context" or rather "the effect of context"?